analytical chemistry/sensory biophysics

aptasensor, ceria, epithelial sodium channel (ENaC), hypertension, biosensor, aptamer

**Author for correspondence:**
Yeni Wahyuni Hartati
e-mail: yeni.w.hartati@unpad.ac.id

This article has been edited by the Royal Society of Chemistry, including the commissioning, peer review process and editorial aspects up to the point of acceptance.

# An aptasensor using ceria electrodeposited-screen-printed carbon electrode for detection of epithelial sodium channel protein as a hypertension biomarker

Yeni Wahyuni Hartati[1], Dina Ratna Komala[1], Diana Hendrati[1], Shabarni Gaffar[1], Ari Hardianto[1], Yulia Sofiatin[2] and Husein Hernandi Bahti[1]

[1]Department of Chemistry, Faculty of Mathematics and Natural Sciences, and [2]Department of Public Health, Faculty of Medicine, Universitas Padjadjaran, Bandung, Indonesia

YWH, 0000-0003-1463-6352; SG, 0000-0002-3659-4774; AH, 0000-0001-6065-5437; YS, 0000-0002-1409-6010

Epithelial sodium channel (ENaC) is a transmembrane protein that has an essential role in maintaining the levels of sodium in blood plasma. A person with a family history of hypertension has a high enough amount of ENaC protein in the kidneys or other organs, so that the ENaC protein acts as a marker that a person is susceptible to hypertension. An aptasensor involves aptamers, which are oligonucleotides that function similar to antibodies, as sensing elements. An electrochemical aptasensor for the detection of ENaC was developed using a screen-printed carbon electrode (SPCE) which was modified by electrodeposition of cerium oxide ($CeO_2$). The aptamer immobilization was via the streptavidin–biotin system. The measurement of changes in current of the active redox $[Fe(CN)_6]^{3-/4-}$ was carried out by differential pulse voltammetry. The surfaces of SPCE and SPCE/$CeO_2$ were characterized using scanning electron microscopy, voltammetry and electrochemical impedance spectroscopy. The Box–Behnken experimental optimization design revealed the streptavidin incubation time, aptamer incubation time and streptavidin concentrations were 30 min, 30 min and 10.8 µg ml$^{-1}$, respectively. Various concentrations of ENaC were used to obtain the linearity range of 0.05–3.0 ng ml$^{-1}$, and the limits of detection and quantification were 0.012 ng ml$^{-1}$ and 0.038 ng ml$^{-1}$, respectively. This aptasensor method has the potential to measure the ENaC protein levels in urine samples as well as to be a point-of-care device.

# 1. Introduction

Hypertension is a condition in which blood pressure increases abnormally, with both systolic and diastolic blood pressure greater than or equal to 130/80 mmHg. In normal circumstances, blood pressures are less than 120/80 mmHg [1]. Sodium salt intake affects hypertension by increasing plasma volume, heart rate and blood pressure [2]. The sodium affects are related to the epithelial sodium channel (ENaC) protein, a transmembrane protein that regulates the reabsorption of sodium ions in several tissues, such as the lungs, intestines and kidneys [3,4]. The ENaC structure consists of four main subunits, which are $\alpha$, $\beta$, $\gamma$ and $\delta$, and each subunit is encoded by four genes: SCNN1A, SCNN1B, SCNN1G and SCNN1D [5,6].

ENaC can be used as a hypertension biomarker for early diagnosis through its interaction with specific antibodies. Determination of the levels of ENaC protein in urine samples based on antibody methods that have been reported includes the enzyme-linked immunosorbent assay method, which showed that ENaC protein could be measured in the supernatant and centrifuged urine sediment [7], and electrochemical immunosensors of the urine samples could measure ENaC without centrifugation and special treatment [8,9]. However, antibody-based methods have some disadvantages, such as being relatively expensive, and they can only bind to large molecules. Therefore, other alternative methods have been developed, such as the determination of ENaC protein levels through its interaction with aptamers.

The discovery of the aptamer single-strand DNA (ssDNA) that specifically binds to the ENaC protein is promising because of its low production costs. An aptamer is a specific RNA or ssDNA oligonucleotide that can interact with a protein. Generally, aptamer base sequences can be obtained through a selection process using an *in vitro* method known as the systematic evolution of ligands by enrichment exponential (SELEX) [10,11]. Aptamers have relatively small sizes compared to antibodies, which are relatively large [12]. Although conceptually simple, SELEX consumes much time, requires large resources and does not always produce aptamers with the desired characteristics [13]. Other methods for obtaining an aptamer have been reported, including kinetics capillary electrophoresis, the equilibrium and non-equilibrium capillary electrophoresis of equilibrium mixtures [14,15] and rational design *in silico* [16,17]. The use of aptamers in aptasensors has been of researchers' great concern because of their synthetic portability and functional design [18]. Functionalization of aptamers with nanomaterial can change its conformation, that will interfere the binding of aptamers. To prevent these disadvantages, the binding of aptamers is carried out with unmodified nanomaterial or with the addition of a linker [19].

In this aptasensor study, the ENaC aptamer was obtained by the virtual screening result of 41 aptamer sequences in the Protein Data Bank, and the *in silico* method was studied and selected that had the most negative bond energy with ENaC. The aptamer *iSpinach* sequence (PDB ID: 5OB3; resolution: 2004 Å; sequence: 69 nts) [20] was a selected aptamer because it produces a bond energy of $-49.46 \text{ kcal mol}^{-1}$ [21]. Furthermore, an ENaC aptamer can be used to detect ENaC protein electrochemically. This electrochemical ENaC aptamer technique can be used as an alternative method because of its high sensitivity, and its operation is quite easy, requiring a relatively small volume of analyte and a simple instrument [22].

Many nanomaterials with unique characteristics have been applied for signal amplification to increase the sensitivity of the electrochemical method. Metal-based and carbon-based nanomaterials have been reported in many studies and recent reviews [18,23–27]. One of them is nanoceria, or cerium oxide ($CeO_2$) nanoparticles, which have unique properties. Nanoceria has the properties of oxygen transfer capability, switchable redox reactivity, large surface area, variations in shape, size, charge, surface reactivity, surface coating and a rigid framework, making $CeO_2$ very suitable for powerful candidates in the development of electrochemical biosensors [28,29]. Mesoporous-hollow $CeO_2$ nanospheres showed a strong interaction with $-NH_2$ to anchor streptavidin and thionine for matrix metalloproteinase 2 biosensors [28]. The mesoporous $CeO_2$ can also form a bridge bond with the carboxyl functional groups of antibodies without the addition of other agents [30].

Recently a modified screen-printed carbon electrode (SPCE) with $CeO_2$ nanoparticles showed low detection limits for the determination of non-steroidal anti-inflammatory drugs diclofenac by cyclic voltammetry and square wave voltammetry [31]. $CeO_2$ nanoparticles have also been used to modify the SPCE for the current detection of codeine, acetaminophen and caffeine in voltammetric measurement [32]. The $CeO_2$-modified SPCE was introduced with biotinylated aptamer through the streptavidin–biotin interaction, which was recently applied in this study to determine the ENaC protein.

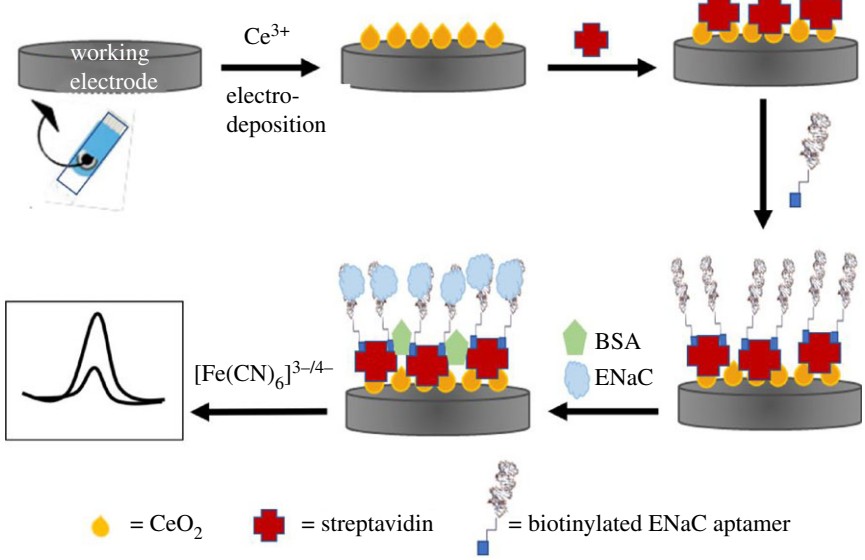

**Figure 1.** Design schematic of the ENaC aptasensor.

In this study, modification of the SPCE performed by electrodeposition of the $Ce^{3+}$ solution of $(Ce(NO_3)_3)$, the immobilization of element sensing via streptavidin-biotinylated aptamer and the detection of ENaC protein cooperating with the differential pulse voltammetry of $[Fe(CN)_6]^{3-/4-}$ are shown to be promising. With this aim, a novel and simple technique can be implemented for the detection of biomarkers of hypertension in urine samples or other clinical fluids.

# 2. Material and methods

## 2.1. Materials

ENaC protein was obtained from Abcam, UK. ENaC aptamer (sequence: 5′–/5Biosg/CGGTGAGGG TCGGGTCCAGTAGGCCTACTGTTGAGTAGTGGGCTCC–3′) was obtained from DT Integrated DNA Technologies, US. Bovine serum albumin (BSA) and potassium ferricyanide ($K_3[Fe(CN)_6]$) were acquired from Sigma Aldrich. Sodium chloride (NaCl), sodium hydroxide (NaOH), phosphate buffer saline (PBS, pH 7.4) and cerium (III) nitrate hexahydrate ($Ce(NO_3)_3 \cdot 6H_2O$) were obtained from Merck. The double-distilled water was produced by PT Ikapharmindo Putramas, Indonesia.

Electrochemical measurements were performed by using a ZP potentiostat with the computer Interface of PSTRACE 5.4 software (Zimmer & Peacock, UK). The SPCE (GS1 Technologies, USA) consisted of carbon working and counter electrodes as well as an Ag/AgCl reference electrode. In addition, other supporting equipment included an autoclave sterilizer (Hirayama Autoclave HVE-50), microtubes and micropipette tips (Eppendorf), magnetic stirrer, mini spin (Eppendorf), hot plate (IKA C-MAG HS 7) and centrifuge (Thermo Scientific MicroCL 17R, USA). The morphologies of the electrode surface were analysed by using scanning electron microscopy (SEM) (Hitachi, Japan).

## 2.2. Methods

The fabrication scheme of ENaC aptasensor with $CeO_2$-modified SPCE is shown in figure 1. The steps of each phase are described below.

## 2.3. Screen-printed carbon electrode surface modification with cerium oxide

The SPCE was dripped with double-distilled water for pre-treatment. Subsequently, 40 µl of $Ce(NO_3)_3$ 100 mg l$^{-1}$ was dropped and electrodeposition was carried out using differential pulse voltammetry (DPV) in the potential range −1.0 to +1.0 V with a scan rate of 0.05 V s$^{-1}$. The electrodes were rinsed carefully with demineralized water and allowed to dry at room temperature. The SPCE modified with

**Table 1.** Factors influencing the experimental conditions in ENaC measurement using the aptasensor.

| | | level | | |
|---|---|---|---|---|
| factors | unit | −1 | 0 | +1 |
| streptavidin incubation time | min | 30 | 60 | 90 |
| streptavidin levels | μg ml$^{-1}$ | 5 | 10 | 15 |
| ENaC aptamer incubation time | min | 10 | 20 | 30 |

cerium was characterized using SEM and cyclic voltammetry. A scan rate of 0.1 V s$^{-1}$ and a potential range between –0.8 and +0.8 V was employed for the cyclic voltammetry analysis under the redox system of $[Fe(CN)_6]^{3-/4-}$. The electrochemical impedance spectroscopy (EIS) measurements were performed using 10 mM of $[Fe(CN)_6]^{4-/3-}$ containing 0.1 M KCl (in 10 mM PBS, pH 7.4). The frequency was set from 5.0 to 500 kHz, and the amplitude of the applied sine wave was 10 mV with the direct current potential set at 0.2 V.

## 2.4. Immobilization of the epithelial sodium channel aptamer on the screen-printed carbon electrode/cerium oxide

In this study, the aptamer immobilization technique was carried out using the streptavidin–biotin system. Streptavidin was immobilized on the SPCE/CeO$_2$ surface by passive adsorption; it was expected that the active group on streptavidin would interact with CeO$_2$ on the surface of the SPCE. The SPCE/CeO$_2$ was rinsed, then 20 μl of streptavidin was dropped onto the electrode surface, and the surface was incubated at 4°C for 30, 60 or 90 min. The streptavidin-modified electrode was then washed three times using pH 7.4 PBS buffer solution. Then, 20 μl of a biotinylated aptamer solution with a concentration of 1 μg ml$^{-1}$ was dropped on each of the SPCEs. The SPCEs were incubated for 10, 20 or 30 min at room temperature and then rinsed with pH 7.4 PBS solution three times.

## 2.5. Determination of aptasensor response to epithelial sodium channel

The part of SPCE that did not bind to streptavidin or the SPCE surface that was not immobilized by streptavidin was blocked using 1% BSA solution and incubated at room temperature for 10 min. The electrodes were then rinsed with pH 7.4 PBS solution. Next, 20 μl of ENaC solution with a certain level was applied on the electrodes and followed by incubation at room temperature for 10 min. The DPV measurement for each concentration was performed by using a 10 mM $[Fe(CN)_6]^{3-/4-}$ solution in 0.1 M KCl. The measurement applied a scanning rate of 0.008 V s$^{-1}$ in the potential range of –1 to +1 V.

## 2.6. Optimization of the experiment parameters

Factors such as streptavidin incubation time ($X1$), streptavidin concentration ($X2$) and ENaC aptamer incubation time ($X3$) were chosen as factors to be optimized in the experiment. Each of these factors was designed through three different levels of the Box–Behnken experimental design, i.e. the lowest (−1), medium (0) and highest (+1) levels as can be observed in table 1. The response of the measurement results from the suggested experiment was then processed and the optimum factor values were analysed by using the MINITAB 18 program.

## 2.7. Calibration curves and limit of detection

The aptasensor response was tested using ENaC with various concentrations of 0.05–3.0 ng ml$^{-1}$. In the range of potential between –0.1 and +1.0 V and a scanning rate of 0.03 V s$^{-1}$, the resulting response was determined by DPV using a redox system of 10 mM $[Fe(CN)_6]^{3-/4-}$ in 0.1 M KCl. The measurements were carried out in the optimal conditions obtained from the Box–Behnken experimental design. The limit of detection (LoD) was determined through the linear curve of concentration versus the difference in peak current average ($\Delta I$).

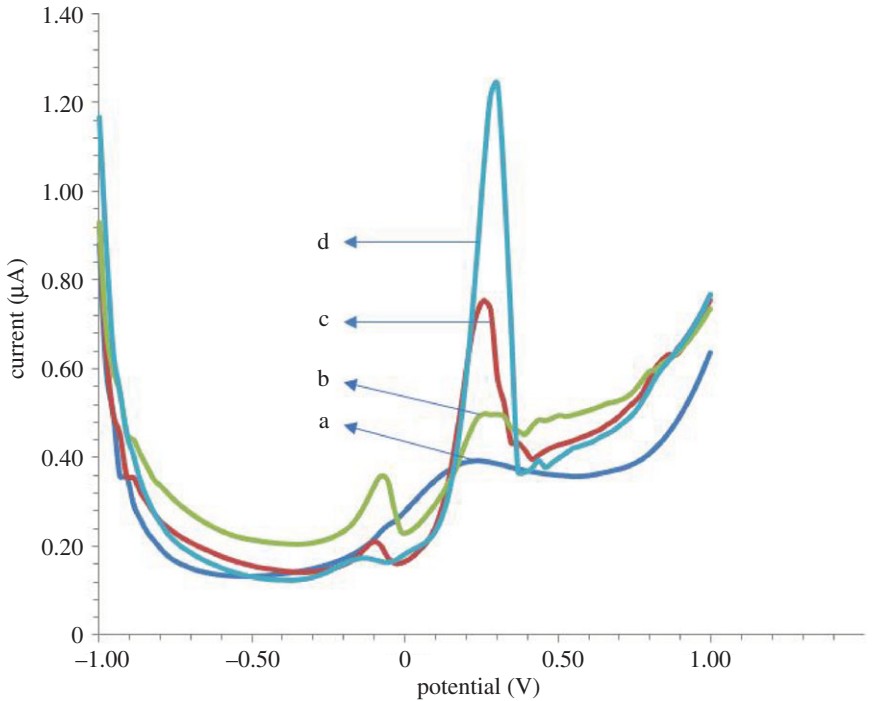

**Figure 2.** Differential pulse voltammogram of various concentration of cerium nitrate; the response was measured using DPV with a scan rate of 0.05 V s$^{-1}$ over a potential range of −1.0 to +1.0 V; the optimized cerium concentrations were (a) 25.0, (b) 50.0, (c) 75.0 and (d) 100.0 µg ml$^{-1}$.

## 2.8. Epithelial sodium channel recovery determination in urine sample matrices

Each 20 µl urine sample was put into four different tubes and 2 µl of ENaC in pH 7.4 PBS solutions with concentrations of 0, 0.75, 1.5 or 3.0 ng ml$^{-1}$ was added to each. The mixtures were then dropped on the electrodes that had been prepared and followed by incubation at room temperature for 10 min. Furthermore, the resulting electrochemical response was determined by DPV. The measurement was performed under a redox system of 10 mM [Fe(CN)$_6$]$^{3-/4-}$ solution in 0.1 M KCl in the range of potential −1 to +1 V and a scanning rate of 0.03 V s$^{-1}$. Triplicate measurements were made at different electrodes. The standard addition method was used to perform a recovery test.

# 3. Results

## 3.1. Screen-printed carbon electrode modification with electrodeposition of cerium oxide

In this study, the cerium nitrate solution was electrodeposited on the SPCE surface. Figure 2 shows the voltammogram of the cerium ion electrodeposition.

Based on the voltammogram in figure 2, the highest peak current of the cerium reduction is indicated by the cerium concentration of 100.0 µg ml$^{-1}$. Furthermore, SPCE modification with Ce(NO$_3$)$_3$ electrodeposition was carried out at a concentration of 100.0 µg ml$^{-1}$.

## 3.2. Characterization of screen-printed carbon electrode/cerium oxide and screen-printed carbon electrode/cerium oxide/aptamer epithelial sodium channel by scanning electron microscopy, differential pulse voltammetry and electrochemical impedance spectroscopy

CeO$_2$-modified SPCE was characterized microscopically by SEM, as shown in figure 3a,b. The SEM results from bare SPCE in figure 3a show that the surface is less homogeneous, whereas in figure 3b, the morphology of SPCE after being modified by CeO$_2$ shows a more homogeneous surface. SPCE/CeO$_2$ and SPCE/CeO$_2$/streptavidin-biotinylated aptamers were characterized by DPV by monitoring changes in the current response of 10 mM [Fe(CN)6]$^{3-/4-}$ in 0.1 M KCl solution in the potential range −1.0 to +1.0 V and a scan rate of 0.008 V s$^{-1}$, as can be seen in figure 3c.

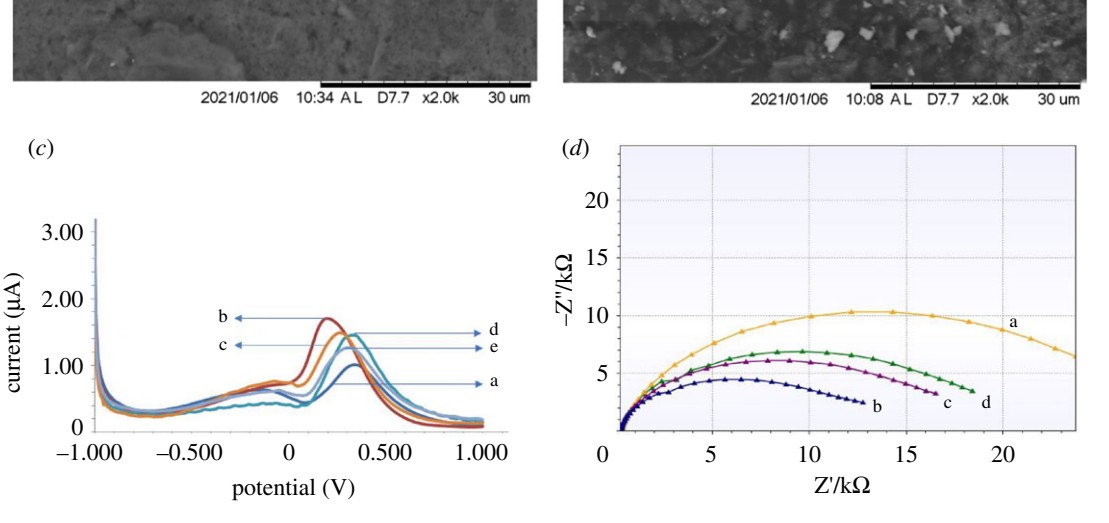

**Figure 3.** (a) Microscopic surface morphology for bare SPCE. (b) SEM morphology of SPCE/CeO$_2$. (c) Differential pulse voltammogram of 10 mM [Fe(CN)$_6$]$^{3-/4-}$ in KCl 0.1 M on: a, bare SPCE; b, SPCE/CeO$_2$; c, SPCE/CeO$_2$/streptavidin; d, SPCE/CeO$_2$/streptavidin/aptamer; and e, SPCE/CeO$_2$/streptavidin/aptamer/ENaC. (d) Nyquist plot of EIS results at a frequency of 100 Hz for: a, bare SPCE; b, SPCE/CeO$_2$; c, SPCE/CeO$_2$/streptavidin; and d, SPCE/CeO$_2$/streptavidin/aptamer.

The morphology of SPCE can be distinguished before and after modification with CeO$_2$, as shown in figure 3a,b. From figure 3c, it can be seen that there is an increase in the peak current of [Fe(CN)$_6$]$^{3-/4-}$ in SPCE that has been modified by CeO$_2$, and this shows that the electrodeposition process had proceeded. The peak of [Fe(CN)$_6$]$^{3-/4-}$ also shifted to the left indicating that CeO$_2$ has high conductivity, which contributes to an easier electron transfer process than in bare SPCE. A decrease in flow occurs when streptavidin sticks (curve c), owing to streptavidin being a non-electroactive macromolecule, so it can block the electron transfer process of [Fe(CN)$_6$]$^{3-/4-}$. At the time of immobilization of the ENaC aptamer (curve d), there was also a decrease in current because the electrode surface was getting denser. This shows that the immobilization was successful so that the reduction current of [Fe(CN)$_6$]$^{3-/4-}$ will be lower and cause the species [Fe(CN)$_6$]$^{3-/4-}$ to be farther from the surface of the electrode as a result. The transfer of electrons is hindered owing to the decrease in CeO$_2$-modified electrode conductivity. The percentage of decrease in peak current was 36.95%.

A 1% BSA solution was used to cover the unoccupied sites on the SPCE surface. The BSA molecule fills in the small gaps in the empty areas at the electrodes. The goal is so the measurement does not produce currents that can interfere with the analysis results of the analyte. Furthermore, to remove the excess ENaC and other molecules, the modified SPCE was rinsed with PBS buffer pH 7.4. The incubation of modified SPCE was carried out at room temperature for 10 min. ENaC protein will bind specifically to ENaC aptamer which has been immobilized on an SPCE. In figure 3c (curve e), it can be seen that the attachment of the ENaC protein to aptamer ENaC causes a decrease in the peak of the [Fe(CN)$_6$]$^{3-/4-}$ current. Because ENaC is a non-electroactive and large biomolecule, it hinders the process of electron transfer on the SPCE surface. The measured current response will be inversely proportional to the number of non-electroactive biomolecules involved (ENaC). The higher the ENaC concentration, the lower the resulting current response will be because the more ENaC that sticks to the SPCE surface can hinder the process of electron transfer from [Fe(CN)$_6$]$^{3-/4-}$. Conversely, the lower the ENaC concentration, the higher the resulting current response will be.

Figure 3*d* shows the modified SPCE surface characterized using EIS. The circle section of the Nyquist plot at higher frequencies corresponds to the electron transfer of $[Fe(CN)_6]^{3-/4-}$. The difference in semicircular diameter or Rct indicates that SPCE has been successfully modified, the higher the concentration of biomolecules, the higher the Rct, inversely proportional to the current in the electron transfer process [33]. In figure 3*d*, spectrum a shows a large Rct for the SPCE before modification. The SPCE modified by $CeO_2$ shows a smaller impedance value as can be seen in spectrum b which means that the resistance is getting smaller; this is inversely proportional to the increased electron transfer current. In spectra c and d, after immobilization of streptavidin and aptamer ENaC, an Rct is also obtained which is smaller than spectrum a, owing to the smaller resistance during the electron transfer process, but the resistance is greater than the ceria-modified SPCE because there are non-electroactive molecules such as streptavidin which blocks the process of electron transfer on the SPCE surface.

## 3.3. Box–Behnken experiment design for experiment optimization

Three factors such as streptavidin incubation time ($X1$), streptavidin concentration ($X2$) and anti-ENaC aptamer incubation time ($X3$) were selected as factors to be optimized in the experiment. The incubation time and the stability of the biomolecules on the electrode surface were the most important parameters. The optimum incubation time determines the time required to bond completely to the electrode surface. When the incubation time is lower, it will result in an imperfect bond. Meanwhile, if the incubation time is too long. This will cause the bonds to become saturated [34]. The effect of streptavidin incubation time, streptavidin concentration and aptamer anti-ENaC incubation time on aptasensor current response was tested with DPV in the potential range –1.0 to 1.0 V with a scanning rate of $0.008 V s^{-1}$ in 10 mM of $[Fe(CN)_6]^{3-/4-}$ containing 0.1 M KCl solution. Streptavidin plays a role in the development of this aptasensor because its function is to direct the orientation of the biotinylated aptamer that is immobilized on the electrode surface.

Streptavidin was dropped onto SPCE/$CeO_2$ with various concentrations of 5, 10 or 15 µg ml$^{-1}$ and incubated with time variations of 30, 60 or 90 min at –4°C. The ENaC aptamer with a concentration of 1 µg ml$^{-1}$ was incubated at room temperature for various times of 10, 20 or 30 min.

The Box–Behnken experimental design was employed to obtain the optimum value of every factor. The experiments using three factors with three different levels were carried out 15 times in a triplicate manner so that there were 30 experiments (data not shown). From each of these experiments, the optimum value of several factors was obtained, including the incubation time for streptavidin of 30 min, the incubation time for aptamer anti-ENaC of 30 min and the streptavidin concentration of 10.76 µg ml$^{-1}$.

## 3.4. Calibration curve

The various ENaC antigen concentrations used were 0.046875, 0.09375, 0.1875; 0.375, 0.75; 1.5 and 3.0 ng ml$^{-1}$. The voltammetric aptasensor responses of the redox system of $[Fe(CN)_6]^{3-/4-}$ solution are shown in figure 4*a*.

The decrease in peak currents was then plotted on a graph against the various ENaC concentrations to create a calibration curve as in figure 4*b*. The linear regression equation $Y = 0.0652X + 0.2073$ with $R^2 = 0.9979$ was obtained. The LoD was evaluated using the blank signal, $y_B$, plus three times standard deviations of the blank, $s_B$. The $y_B$ is the intercept of the calibration curve, and the $s_B$ was obtained from the random errors in the $y$-direction. The limit of quantification (LoQ) was evaluated using 10 times standard deviation of the blank [35]. The LoD was found at the ENaC level of 0.012 ng ml$^{-1}$, and the LoQ was 0.038 ng ml$^{-1}$.

## 3.5. Determination of epithelial sodium channel levels in urine samples

A recovery test was carried out to evaluate the consistency of the new method, concerning its precision. The standard addition method was used for the determination of sample concentrations of non-hypertensive individuals. The spikes of 0.75, 1.5 and 3.0 ng ml$^{-1}$ were added to the urine sample, and a standard addition curve was obtained as the voltammetric response with a linear regression equation. The ENaC level in the urine sample as the $X$-intercept was 0.842 ng ml$^{-1}$. The recoveries of triplicate measurement of ENaC concentrations were obtained, as shown in table 2.

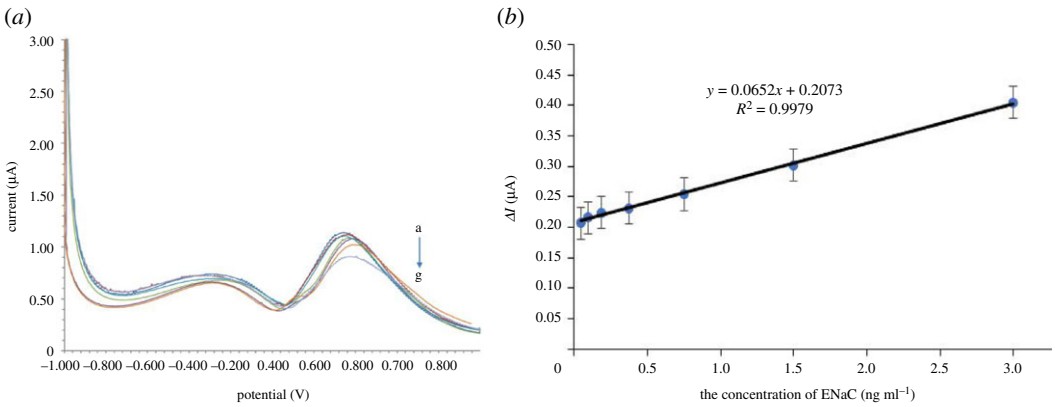

**Figure 4.** (*a*) The decrease of peak current in voltammograms for determination of each ENaC concentration a–g (0.046875, 0.09375, 0.1875; 0.375, 0.75; 1.5 and 3.0 ng ml$^{-1}$) using 10 mM of $[Fe(CN)_6]^{3-/4-}$ in 0.1 M KCl. (*b*) Plot of current to concentration.

**Table 2.** The recovery of the method.

| no. | spike (ng ml$^{-1}$) | recovery (%) |
|---|---|---|
| 1 | 0.0682 | $96.2 \pm 1.03$ |
| 2 | 0.1364 | $97.7 \pm 1.16$ |
| 3 | 0.2727 | $107.5 \pm 0.03$ |

Based on table 2, the recoveries were in the 90–110% range; therefore, the resulting recovery figures verified the precision of the new method.

## 4. Discussion

Ceria has attracted great attention in sensor technology, owing to its unique characteristics such as good mechanical stability, high electronic conductivity, non-toxicity and biocompatibility [36,37]. In the development of modification of SPCE with $CeO_2$ usually, cerium is made in the form of an oxide with a nanometre size of the precursor cerium nitrate $(Ce(NO_3)_3)$ or other cerium salts. The excellent conductance properties of cerium play an essential role in the load transfer process [32].

The 40 µl of 100 µg ml$^{-1}$ cerium nitrate solution was used to modify the SPCE. The electrodeposition reaction process of cerium on the SPCE surface was carried out in an open circuit so that the possible reaction that occurs is the oxidation of $Ce^{3+}$ or Ce (III) ions in $Ce(NO_3)_3$ solution by $O_2$ or air to become $Ce^{4+}$ or Ce (IV). This cathodic electrodeposition releases one electron, with the steps of the $Ce^{3+}$ oxidation reaction to $Ce^{4+}$ being as follows [38]:

$$4Ce^{3+} + O_2 + 14H_2O \rightarrow 4Ce(OH)_4 + 12H^+,$$

the reduction of $O_2$, $H_2O$ and $H_3O^+$ are as follows:

$$2H_3O^+ + 2e^- \rightarrow H_2 + 2H_2O,$$
$$2H_2O + 2e^- \rightarrow H_2 + 2OH^-$$

and

$$O_2 + 2H_2O + 4e^- \rightarrow 4OH^-,$$

the consumption reaction of $H_3O^+$ and production of $OH^-$ on the formation of $Ce(OH)_3$ or $Ce(OH)_2^{2+}$ are as follows:

$$Ce^{3+} + 3OH^- \rightarrow Ce(OH)_3$$

**Table 3.** The electrochemical determination of ENaC proteins.

| no. | methods | results | reference |
|---|---|---|---|
| 1 | electrochemical immunosensor utilising SPCE/rGO/Anti-ENaC | linear range: 0.01–1.5 ng ml$^{-1}$ <br> LoD: 0.198 ng ml$^{-1}$ | [8] |
| 2 | electrochemical immunosensor using SPCE/Au/ Anti-ENaC-AuNp bioconjugate | linear range: $9.375 \times 10^{-2}$ to 1.0 ng ml$^{-1}$ <br> LoD: 0.084 ng ml$^{-1}$ | [9] |
| 3 | electrochemical aptasensor using SPCE/CeO$_2$/SVbiotinylated aptamer of ENaC protein | linear range: 0.05–3.0 ng ml$^{-1}$ <br> LoD: 0.012 ng ml$^{-1}$ | this work |

and

$$4Ce^{3+} + O_2 + 4OH^- + 2H_2O \rightarrow 4Ce(OH)_2^{2+}.$$

Oxidation of Ce$^{3+}$ to Ce$^{4+}$:

$$Ce(OH)_3 \rightarrow CeO_2 + H_3O^+ + e^-,$$
$$Ce(OH)_2^{2+} \rightarrow CeO_2 + 2H_2O.$$

Based on the peak current that appears in figure 2, there is an electrogenerated peak current of more than one peak at different potentials, possibly because of the reaction stages described above. The nitrate reduction and the hydrogen evolution reactions for the first period of deposition may become predominant [38].

CeO$_2$ as a source of lanthanide ions is a strong Lewis base. This causes it to be a strong ligand against phosphate and carboxyl groups. CeO$_2$ has also been reported to be desirable as a supporting material to immobilize biomolecules. CeO$_2$ is slightly negatively charged at neutral pH with $\zeta$-potential [29,39]. In this study, SPCE/CeO$_2$ was modified with streptavidin, owing to CeO$_2$ having a high affinity for biomolecules through electrostatic interaction with the –NH$_2$ group of streptavidin. Thus, it becomes a strong anchor for the binding of the biotinylated aptamer and has a directional orientation when the aptamer immobilizes on the electrode surface. The interaction between biotin and streptavidin has a high-affinity bond with an association constant of $K_a$=1015 M$^{-1}$ [40]. The aptamer is immobilized on the electrode surface through a non-covalent interaction between biotin and streptavidin so that it can block the process of electron transfer on the electrode surface.

A comparison of the detection limits for the electrochemical determination of ENaC proteins is shown in table 3.

# 5. Conclusion

The aptasensor developed in this study was used to measure ENaC levels in urine samples. A person with a family history of hypertension is considered to have the disease if the ENaC content in his urine is more than 2.7 ng ml$^{-1}$. For a person from a family with hypertension, the threshold is more than 4.0 ng ml$^{-1}$, while non-hypertensive is above 1.12 ng ml$^{-1}$ [7].

Data accessibility. The datasets supporting this article have been uploaded as part of the electronic supplementary material.

Authors' contributions. Y.W.H. and Y.S.: substantial contributions to conception and design, or acquisition of data, or analysis and interpretation of data; D.R.K., D.H. and A.H.: drafting the article or revising it critically for important intellectual content; S.G., Y.W.H. and H.H.B.: final approval of the version to be published; and D.R.K. and Y.W.H.: agreement to be accountable for all aspects of the work in ensuring that questions related to the accuracy or integrity of any part of the work are appropriately investigated and resolved.

Competing interests. We have no competing interests.

Funding. This work was funded by the PDUPT Scheme Research of Indonesian Ministry of Research, Technology and the National Innovation Agency no. 1827/UN6.3.1/LT/2020, and Universitas Padjadjaran Academic Leadership grant no. 1427/UN6.3.1/LT/2020.

Acknowledgements. Thanks to all members of the biosensor team who have collaborated to facilitate this research in the biosensor lab of the Department of Chemistry, Universitas Padjadjaran.

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
