## [Peer Review File · Royal Society Open Science]

Review History

RSOS-202040.R0 (Original submission)

Review form: Reviewer 1 (Dianping Tang)

Is the manuscript scientifically sound in its present form?

No

Are the interpretations and conclusions justified by the results?

Yes

Is the language acceptable?

Yes

Do you have any ethical concerns with this paper?

Yes

Have you any concerns about statistical analyses in this paper?

Yes

Recommendation?

Major revision is needed (please make suggestions in comments)

Comments to the Author(s)

This manuscript designs an aptasensing platform for the detection of epithelial sodium channel protein by using ceria electrodeposited screen-printed carbon electrode. The corresponding aptamers were immobilized on the electrodeposited electrode by coupling with biotin-avidin reaction. The electrochemical signal was acquired on the basis of the change before and after target-aptamer reaction. However, modification is requested before being considered for publication in this journal.

Specific comments:

1. Recently, different methods and strategies have been reported and designed for the quantitative monitoring of disease-relative biomarkers. What are the advantages and disadvantages of the developed electrochemical aptasensor in comparison with conventional detection systems? Actually, they should be simply described in the main text.
2. In Figure 2, differential pulse voltammograms (DPV) were determined toward different-concentration cerium nitrate. However, the potentials are different toward different-concentration cerium nitrate. Please explain possible reason! In addition, please remark the meaning of letters "a,b,c,d" in Figure 2.
3. Generally, the Nyquist diagrams involve in the bulk and surface impedance in the high-frequency region in addition to Warburg impedance in the low-frequency region. However, only bulk impedance was observed in the high-frequency region. Why? Please explain the possible reason.
4. Many nanomaterials with unique characteristics have been applied for signal amplification to increase the sensitivity of the electrochemical method. Please provide the relative works on the description of nanomaterials (e.g., *Anal. Chem.* 2020, 92:363). Further, the advantages of aptamers should be further discussed in the introduction because this study mainly focused on the aptasensors (Please refer to the relative works, e.g., *Anal. Chem.* 2019, 91:1260; *Chem. Commun.* 2018, 54:7199; *Anal. Chem.* 2020, 92:1470; *ACS Sens.* 2018, 3:632). Moreover, literatures are insufficient.
5. How to evaluate the accuracy of the electrochemical aptasensor for the determination of ENaC levels in urine samples? As a newly developed detection method, the method accuracy is very important. In addition, the mechanism of electrochemical signal should be further discussed in the main text.

Review form: Reviewer 2

Is the manuscript scientifically sound in its present form?

Yes

Are the interpretations and conclusions justified by the results?

Yes

Is the language acceptable?

No

Do you have any ethical concerns with this paper?

No

Have you any concerns about statistical analyses in this paper?

No

Recommendation?

Accept with minor revision (please list in comments)

Comments to the Author(s)

Manuscript ID: RSOS-202040

Title: An Aptasensor Using Ceria Electrodeposited-SPCE For Detection of Epithelial Sodium Channel Protein As A Hypertension Biomarker

Recommendation: Minor Revision

Comments:

In this manuscript, an electrochemical aptasensor was developed to detect ENaC utilizing a cerium oxide modified screen-printed carbon electrode (SPCE) using the DPV technique. The aptamer immobilization was performed via the streptavidin-biotin system. The present work is interesting and the manuscript is also well organized. It can be published after some revisions.

1. Abstract or summary is the most important part of an article. It should be precise and easy to understand. Here it seems perfect length with sufficient information, but the meaning of the following quoted sentence is not clear. Please rewrite this sentence. "An aptasensor has involved the aptamers as the element sensing that is the oligonucleotide that functions similarly to an antibody". Also, remove the grammatical errors from the following sentences "The various concentration of ENaC was used to obtain the linearity range of 0.05 to 3.0 ng/mL, limit of detection and quantification were 0.012 ng/mL and 0.038 ng/mL, respectively" and "The aptamer immobilization via the streptavidin-biotin system".

2. Introduction is well written. Still few things need to fix.

Add appropriate reference to "Sodium salt intake affects hypertension by increasing plasma volume, heart rate, and blood pressure" and "Recently modified SPCE with CeO₂ nanoparticles showed low detection limits for the determination of non steroidal anti-inflammatory drugs diclofenac by cyclic voltammetry and square wave voltammetry." Define ssDNA as it comes for the first time in the manuscript.

What is DPV (DPV) measurement? Implemented should be in sentence case.

3. Figure 1 is missing!

4. Table 1: factor name 'concentration' should be in sentence case.

5. Pg 6 line 59: SPCE was modified with cerium or cerium oxide?

Page 6, line 4-6: make uniform timing presentation 30, 60, and 90 minutes or 10 minutes, 20 minutes, and 30 minutes. Mention PBS concentration. 20 μ L biotinylated aptamer solution added to which electrode? SPCE or modified SPCE?

6. In Fig. 2 caption, what is VPD? Adjust Y-axis value in Fig. 3d to decrease the space.

7. Authors need to show and describe properly the SEM images of modified and unmodified electrodes. In its present condition, neither in SEM images nor in the description no significant difference was observed.

8. In Fig. 4b, the calibration equation and calibration curve don't match with each other. Y-intercept should be ~ 0.18 according to the curve, but in the equation it is 0.2073. Please double-check these slope and intercept values and also include units in the equation.
9. Authors need to show in the revised Manuscript how they have calculated LOD and LOQ.
10. In Table 2, 107.5% recovery is high enough, please double-check this matter.
11. Correct all formulas throughout the Manuscript, like CeO_2 in pg 11 line 18, 2e- in pg 10 line 48-50, etc.
12. Conclusion is too short!

Decision letter (RSOS-202040.R0)

This year has been very difficult for everyone, and we want to take the opportunity to thank you for your continued support in 2020.

The Royal Society Open Science editorial office will be closed from the evening of Friday 18 December 2020 until Monday 4 January 2021. We will not be responding during this time. If you have received a deadline within this time period, please contact us as soon as possible to allow us to extend the deadline. If you receive any automated messages during this time asking you to meet a deadline, we offer apologies and invite you to respond after the festive period or during normal working hours.

With our best for a peaceful festive period and New Year, and we look forward to working with you in 2021.

Dear Dr Hartati:

Title: An Aptasensor Using Ceria Electrodeposited-SPCE For Detection of Epithelial Sodium Channel Protein As A Hypertension Biomarker
Manuscript ID: RSOS-202040

The editor assigned to your manuscript has now received comments from reviewers. We would like you to revise your paper in accordance with the referee and Subject Editor suggestions which can be found below (not including confidential reports to the Editor). Please note this decision does not guarantee eventual acceptance.

Please submit your revised paper before 15-Jan-2021. Please note that the revision deadline will expire at 00.00am on this date. If we do not hear from you within this time then it will be assumed that the paper has been withdrawn. In exceptional circumstances, extensions may be possible if agreed with the Editorial Office in advance. We do not allow multiple rounds of revision so we urge you to make every effort to fully address all of the comments at this stage. If deemed necessary by the Editors, your manuscript will be sent back to one or more of the original reviewers for assessment. If the original reviewers are not available we may invite new reviewers.

RSC Associate Editor:
Comments to the Author:
(There are no comments.)

RSC Subject Editor:
Comments to the Author:
(There are no comments.)

Reviewers' Comments to Author:
Reviewer: 1

Comments to the Author(s)

This manuscript designs an aptasensing platform for the detection of epithelial sodium channel protein by using ceria electrodeposited screen-printed carbon electrode. The corresponding aptamers were immobilized on the electrodeposited electrode by coupling with biotin-avidin reaction. The electrochemical signal was acquired on the basis of the change before and after target-aptamer reaction. However, modification is requested before being considered for publication in this journal.

Specific comments:

1. Recently, different methods and strategies have been reported and designed for the quantitative monitoring of disease-relative biomarkers. What are the advantages and disadvantages of the developed electrochemical aptasensor in comparison with conventional detection systems? Actually, they should be simply described in the main text.
2. In Figure 2, differential pulse voltammograms (DPV) were determined toward different-concentration cerium nitrate. However, the potentials are different toward different-concentration cerium nitrate. Please explain possible reason! In addition, please remark the meaning of letters "a,b,c,d" in Figure 2.
3. Generally, the Nyquist diagrams involve in the bulk and surface impedance in the high-frequency region in addition to Warburg impedance in the low-frequency region. However, only bulk impedance was observed in the high-frequency region. Why? Please explain the possible reason.
4. Many nanomaterials with unique characteristics have been applied for signal amplification to increase the sensitivity of the electrochemical method. Please provide the relative works on the description of nanomaterials (e.g., Anal. Chem. 2020, 92:363). Further, the advantages of aptamers should be further discussed in the introduction because this study mainly focused on the aptasensors (Please refer to the relative works, e.g., Anal. Chem. 2019, 91:1260; Chem. Commun. 2018, 54:7199; Anal. Chem. 2020, 92:1470; ACS Sens. 2018, 3:632). Moreover, literatures are insufficient.
5. How to evaluate the accuracy of the electrochemical aptasensor for the determination of ENaC levels in urine samples? As a newly developed detection method, the method accuracy is very important. In addition, the mechanism of electrochemical signal should be further discussed in the main text.

Reviewer: 2

Comments to the Author(s)

Manuscript ID: RSOS-202040

Title: An Aptasensor Using Ceria Electrodeposited-SPCE For Detection of Epithelial Sodium Channel Protein As A Hypertension Biomarker

Recommendation: Minor Revision

Comments:

In this manuscript, an electrochemical aptasensor was developed to detect ENaC utilizing a cerium oxide modified screen-printed carbon electrode (SPCE) using the DPV technique. The aptamer immobilization was performed via the streptavidin-biotin system. The present work is interesting and the manuscript is also well organized. It can be published after some revisions.

1. Abstract or summary is the most important part of an article. It should be precise and easy to understand. Here it seems perfect length with sufficient information, but the meaning of the following quoted sentence is not clear. Please rewrite this sentence. "An aptasensor has involved the aptamers as the element sensing that is the oligonucleotide that functions similarly to an antibody". Also, remove the grammatical errors from the following sentences "The various concentration of ENaC was used to obtain the linearity range of 0.05 to 3.0 ng/mL, limit of detection and quantification were 0.012 ng/mL and 0.038 ng/mL, respectively" and "The aptamer immobilization via the streptavidin-biotin system".

2. Introduction is well written. Still few things need to fix.

Add appropriate reference to "Sodium salt intake affects hypertension by increasing plasma volume, heart rate, and blood pressure" and "Recently modified SPCE with CeO₂ nanoparticles

showed low detection limits for the determination of non steroidal anti-inflammatory drugs diclofenac by cyclic voltammetry and square wave voltammetry.” Define ssDNA as it comes for the first time in the manuscript.

What is DPV (DPV) measurement? Implemented should be in sentence case.

3. Figure 1 is missing!

4. Table 1: factor name ‘concentration’ should be in sentence case.

5. Pg 6 line 59: SPCE was modified with cerium or cerium oxide?

Page 6, line 4-6: make uniform timing presentation 30, 60, and 90 minutes or 10 minutes, 20 minutes, and 30 minutes. Mention PBS concentration. 20 μ L biotinylated aptamer solution added to which electrode? SPCE or modified SPCE?

6. In Fig. 2 caption, what is VPD? Adjust Y-axis value in Fig. 3d to decrease the space.

7. Authors need to show and describe properly the SEM images of modified and unmodified electrodes. In its present condition, neither in SEM images nor in the description no significant difference was observed.

8. In Fig. 4b, the calibration equation and calibration curve don’t match with each other. Y-intercept should be ~ 0.18 according to the curve, but in the equation it is 0.2073. Please double-check these slope and intercept values and also include units in the equation.

9. Authors need to show in the revised Manuscript how they have calculated LOD and LOQ.

10. In Table 2, 107.5% recovery is high enough, please double-check this matter.

11. Correct all formulas throughout the Manuscript, like CeO_2 in pg 11 line 18, 2e- in pg 10 line 48-50, etc.

12. Conclusion is too short!

Author's Response to Decision Letter for (RSOS-202040.R0)

See Appendix A.

Decision letter (RSOS-202040.R1)

Dear Dr Hartati:

Title: An Aptasensor Using Ceria Electrodeposited-SPCE for Detection of Epithelial Sodium Channel Protein as a Hypertension Biomarker

Manuscript ID: RSOS-202040.R1

It is a pleasure to accept your manuscript in its current form for publication in Royal Society Open Science. The chemistry content of Royal Society Open Science is published in collaboration with the Royal Society of Chemistry.

RSC Associate Editor
Comments to the Author:
(There are no comments.)

Reviewer(s)' Comments to Author:

Appendix A

Comments from the reviewers and authors

Reviewer 1:

Comments to the Author(s)

This manuscript designs an aptasensing platform for the detection of epithelial sodium channel protein by using ceria electrodeposited screen-printed carbon electrode. The corresponding aptamers were immobilized on the electrodeposited electrode by coupling with biotin-avidin reaction. The electrochemical signal was acquired on the basis of the change before and after target-aptamer reaction. However, modification is requested before being considered for publication in this journal.

Specific comments:

No	Comments from the editors and reviewers	Comments from author
1	Recently, different methods and strategies have been reported and designed for the quantitative monitoring of disease-relative biomarkers. What are the advantages and disadvantages of the developed electrochemical aptasensor in comparison with conventional detection systems? Actually, they should be simply described in the main text.	Thank you very much for the advice, the advantages and disadvantages of the electrochemical aptasensor has been added to the introduction, highlight with blue letters (ref no. 18 and 19)
2	In Figure 2, differential pulse voltammograms (DPV) were determined toward different-concentration cerium nitrate. However, the potentials are different toward different-concentration cerium nitrate. Please explain possible reason! In addition, please remark the meaning of letters "a,b,c,d" in Figure 2.	Based on Figure 2, there is more than one electrogenerated peak current that appears at a different potential, this probably due to the reaction steps described previously. The nitrate reduction and the hydrogen evolution reactions for the first period of deposition may become predominant. The explanation has been added in the discussion section, highlight with blue letters. The caption of Figure 2 has been revised; The description of a,b,c,d was missed, and have been revised.
3	Generally, the Nyquist diagrams involve in the bulk and surface impedance in the high-frequency region in addition to Warburg impedance in the low-frequency region. However, only bulk impedance was	The region of frequency used in the experiment was 0,5 Hz to 500KHz. The Nyquist Plot for a simplified Randles cell is always a semicircle. We assumed that this information is sufficient to see

	observed in the high-frequency region. Why? Please explain the possible reason.	the difference of R_{ct} , therefore impedance in the low-frequency region was not done.
4	Many nanomaterials with unique characteristics have been applied for signal amplification to increase the sensitivity of the electrochemical method. Please provide the relative works on the description of nanomaterials (e.g., Anal. Chem. 2020, 92:363). Further, the advantages of aptamers should be further discussed in the introduction because this study mainly focused on the aptasensors (Please refer to the relative works, e.g., Anal. Chem. 2019, 91:1260; Chem. Commun. 2018, 54:7199; Anal. Chem. 2020, 92:1470; ACS Sens. 2018, 3:632). Moreover, literatures are insufficient.	Some new related references have been added and cited.
5	How to evaluate the accuracy of the electrochemical aptasensor for the determination of ENaC levels in urine samples? As a newly developed detection method, the method accuracy is very important. In addition, the mechanism of electrochemical signal should be further discussed in the main text.	The accuracy of the electrochemical aptasensor was done by measuring the concentration of standard ENaC protein recombinant as standard material, and the accuracy is 99,9% (supplementary information). Based on the AOAC, if the standard CRM is not available, the accuracy can be measured by spike recovery to the urine samples, and the range that can be tolerated is between 90-110%.

Reviewer: 2

Comments to the Author(s)
Manuscript ID: RSOS-202040

Title: An Aptasensor Using Ceria Electrodeposited-SPCE For Detection of Epithelial Sodium Channel Protein As A Hypertension Biomarker

Recommendation: Minor Revision

Comments:

In this manuscript, an electrochemical aptasensor was developed to detect ENaC utilizing a cerium oxide modified screen-printed carbon electrode (SPCE) using the DPV technique. The aptamer immobilization was performed via the streptavidin-biotin system. The present work is interesting and the manuscript is also well organized. It can

be published after some revisions.

No	Comments from the editors and reviewers	Comments from author
1	Abstract or summary is the most important part of an article. It should be precise and easy to understand. Here it seems perfect length with sufficient information, but the meaning of the following quoted sentence is not clear. Please rewrite this sentence. “An aptasensor has involved the aptamers as the element sensing that is the oligonucleotide that functions similarly to an antibody”. Also, remove the grammatical errors from the following sentences “The various concentration of ENaC was used to obtain the linearity range of 0.05 to 3.0 ng/mL, limit of detection and quantification were 0.012 ng/mL and 0.038 ng/mL, respectively” and “The aptamer immobilization via the streptavidin-biotin system”.	Thank you very much for your comment and valuable advice on the manuscript. The English language has been improved. We revised some of the sentences. Sentences in abstract and summary have been re-written and grammatical errors have been revised.
2	Introduction is well written. Still few things need to fix. Add appropriate reference to “Sodium salt intake affects hypertension by increasing plasma volume, heart rate, and blood pressure” and “Recently modified SPCE with CeO₂ nanoparticles showed low detection limits for the determination of non-steroidal anti-inflammatory drugs diclofenac by cyclic voltammetry and square wave voltammetry.” Define ssDNA as it comes for the first time in the manuscript. What is DPV (DPV) measurement? Implemented should be in sentence case.	The references have been added to each cited sentence. ssDNA it comes for the first time in the manuscript has defined (highlight with green letters). DPV-differential pulse voltammetry measurement has been implemented in sentences (green letters).
3	Figure 1 is missing!	Figure 1 has been added
4	Table 1: factor name ‘concentration’ should be in sentence case.	Factor name ‘Concentration’ is replaced with “level”
5	Pg 6 line 59: SPCE was modified with cerium or cerium oxide? Page 6, line 4-6: make uniform timing presentation 30, 60, and 90 minutes or 10 minutes, 20 minutes, and 30 minutes.	Page 6: It should be ‘SPCE was modified with cerium oxide, via electrodeposition of cerium nitrate solution. The sentences have revised (green letters)

	Mention PBS concentration. 20 μ L biotinylated aptamer solution added to which electrode? SPCE or modified SPCE?	Timing presentation in Box-Benken design are different; 30,60, and 90 minutes are times for modification SPCE with streptavidin, while 10, 20, and 30 minutes are aptamer-biomarker time interaction. The concentration of PBS was 10mM. And biotinylated aptamer solution added to modified SPCE (SPCE/CeO ₂ /Streptavidin) All related sentences have been revised.
6	In Fig. 2 caption, what is VPD? Adjust Y-axis value in Fig. 3d to decrease the space.	The caption of Fig 2 has been revised, which is DPV (typo). Fig 3D has been revised.
7	Authors need to show and describe properly the SEM images of modified and unmodified electrodes. In its present condition, neither in SEM images nor in the description no significant difference was observed.	The analysis by SEM has been re-done. Now the difference between SPCE and modified SPCE can be seen clearly.
8	In Fig. 4b, the calibration equation and calibration curve don't match with each other. Y-intercept should be \sim 0.18 according to the curve, but in the equation it is 0.2073. Please double-check these slope and intercept values and also include units in the equation.	Thank you very much. Fig 4b probably was stretched when edited the space of the page, therefore the numbers were shifted. Figure 4b has been revised
9	Authors need to show in the revised Manuscript how they have calculated LOD and LOQ.	LoD and LoQ were measured by the equation that has been added to the manuscript.
10	In Table 2, 107.5% recovery is high enough, please double-check this matter.	The recovery was measured based on AOAC, the accepted value is 10% above and below 100%. (90-110%). Therefore, 107,5% is still acceptable.
11	Correct all formulas throughout the Manuscript, like CeO ₂ in pg 11 line 18, 2e- in pg 10 line 48-50, etc.	The formulas have been revised.
12	Conclusion is too short!	The conclusion has been elaborated.